# Evaluation and Selection of New *Trichogramma* spp. as Biological Control Agents of the Guatemalan Potato Moth (*Tecia solanivora*) in Europe

**DOI:** 10.3390/insects14080679

**Published:** 2023-08-01

**Authors:** Jorge Gavara, Tomás Cabello, Manuel Gámez, Saskia Bastin, Estrella Hernández-Suárez, Ana Piedra-Buena

**Affiliations:** 1Instituto Canario de Investigaciones Agrarias (ICIA), Finca Isamar, Ctra. de El Boquerón, s/n, 38201 San Cristóbal de la Laguna, Spain; bastin.saskia@hotmail.be (S.B.); ehernand@icia.es (E.H.-S.); apbuena@icia.es (A.P.-B.); 2Department d’Ecosistemes Agroforestals, Universitat Politècnica de València, Camino de Vera, s/n, 46022 València, Spain; 3Research Centre for Mediterranean Intensive Agrosystems and Agri-Food Biotechnology (CIAMBITAL), Universidad de Almería (UAL), Ctra. de Sacramento, s/n, 04120 Cañada de San Urbano (La), Spain; tcabello@ual.es (T.C.); mgamez@ual.es (M.G.)

**Keywords:** biological control, crop protection, egg parasitism, semi-field, host suitability, parasitoids, storage

## Abstract

**Simple Summary:**

The Guatemalan potato moth, *Tecia solanivora*, is an invasive pest that has spread to several countries of South America and reached Spain, arriving in Tenerife in 1999, now representing the most important pest in potato crops in the Canary Islands, where it causes significant crop losses, both in the field and in storage. In the field, the use of phytosanitary treatments to manage Guatemalan potato moth populations has been ineffective, and chemical control in warehouses has limited potential, as most insecticides cannot be applied to crops shortly before they are marketed, in addition to the current lack of authorized chemical product treatments. In this context, the search for biological control agents for Guatemalan potato moth pest has become especially important. In this sense, the evaluation of the potential use of *Trichogramma* species is of interest because egg parasitoids kill the host before hatching, resulting in a direct reduction in damage to the crop and harvest. In the present work, two species (*T. achaeae* and *T. euproctidis*) were evaluated under laboratory and semi-field conditions. *T. achaeae* was found to have potential as a biocontrol agent under field conditions, and *T. euproctidis* was found to be suitable for use under storage conditions.

**Abstract:**

The Guatemalan potato moth (*Tecia solanivora*) is designated a quarantine pest by the European Union, causing severe production losses in potato crops. No effective chemical control alternatives are currently available, and cultural techniques are unable to reduce harvest losses to acceptable levels. With a focus on biological control, two egg parasitoids (*Trichogramma euproctidis* and *Trichogramma achaeae*) were selected and evaluated for use under field and storage conditions. Laboratory assays (choice and no-choice) indicated the preference of both parasitoids for *T. solanivora* vs. *Phthorimaea operculella*. *Trichogramma euproctidis* showed the highest parasitic activity for both moths. Analysis of functional response (at 15, 20, 25 and 27 °C) confirmed the high parasitic potential of *T. euproctidis*. Furthermore, in assays conducted under darkness conditions, *T. achaeae* was unable to parasitize eggs. However, in semi-field assays, *T. achaeae* was more efficient in searching for eggs in the soil than *T. euproctidis*. Based on these results, *T. achaeae* was selected to be tested under field conditions, and *T. euproctidis* was selected for testing under storage conditions.

## 1. Introduction

*Tecia solanivora* (Povolný, 1973) (Lepidoptera: Gelechiidae), commonly known as the Guatemalan potato moth, is native to Guatemala and was reported for the first time in 1956 [1]. This pest spread to several countries of South America both through trade and the illegal transport of seeds and tubers. In 1999, the Guatemalan potato moth reached Tenerife (Canary Islands, Spain) from Venezuela, then presumably spread from the Canary Islands to mainland Spain: Galicia (2015) and Asturias (2017) [2,3]. The Guatemalan potato moth pest causes important harvest losses in all countries where it is present. If not managed, 50% or more of the harvest may be lost in the field, and up to 100% may be lost in storage facilities [4,5]. The European Union declared the Guatemalan potato moth a quarantine pest owing to its severity, and the import of potato tubers from countries or internal regions with *T. solanivora* presence is forbidden [1]. Female Guatemalan potato moths lay their eggs in soil irregularities and cracks near the stems of plants. When the eggs hatch, the larvae move to the tubers and start to feed on them [6,7]. They remain inside the tubers, feeding and growing until the last instar, when they move toward the soil surface to pupae. Only adult moths live outside of the soil to reproduce and start a new cycle [3]. The process described above necessitates a complex pest management regime. Chemical control is ineffective because only adult moths and the most exposed eggs are removed, whereas most eggs and larvae are protected under the soil. Consequently, current management is based on cultural techniques throughout the growing cycle and on pest population monitoring using pheromone traps [5,8]. However, these techniques are not effective enough to reduce harvest losses to acceptable economic levels for growers either in the field or under storage conditions, as no authorized chemical treatments are currently available, and infested potatoes usually evade phytosanitary visual control due to their undamaged appearance during the first stage of infestation [9,10].

The ineffectiveness of current phytosanitary and cultural techniques to control the Guatemalan potato moth has resulted in an increase in efforts to search for and evaluate potential candidates to improve integrated management of the pest [11,12,13,14,15]. In this sense, the study of egg predators or parasitoids is especially interesting with respect to the biological control of *T. solanivora*, as once the eggs hatch, the larvae migrate into the soil, making it difficult to remove them before they enter the tubers. In particular, this type of biocontrol agent is interesting for storage conditions, where the eggs are laid on the surface of potatoes. Few eggs parasitoids are found on *T. solanivora*; one example is the braconid, *Apanteles* sp., although it does not protect tubers, as this parasitoid does not prevent the growth and feeding of larvae but may reduce the population of the pest in the next generation [12,16,17]. The encyrtid (*Copidosoma koehleri*) (Blanchard, 1940) presents the same limitation, despite sometime proving successful under field conditions on common tuber moth, *Phthorimaea operculella* (Zeller, 1873). Therefore, *C. koehleri* is considered to be among the main natural enemies of potato moths in tuber complexes (*Symmetrischema tangolias* (Gyen, 1913), *P. operculella* and *T. solanivora*) [18]. Several authors have tested this parasitoid against Guatemalan potato moth under laboratory conditions and found that it does not achieve satisfactory acceptance and preference results [19,20]. Ichneumonidae (*Pimpla* sp.) were also found on *T. solanivora* but were discarded because the establishment of rearing was not possible owing to a considerable lack of female offspring [21]. A Colombian trichogrammatid strain, *Trichogramma lopezandinensis* (Sarmiento, 1993), has shown promising results as a biological control agent under semi-storage conditions [22]; however, this species has a restricted distribution and is not present in the Canary Islands nor in mainland Spain [23]. The species of the genus *Trichogramma* (Hymenoptera: Trichogrammatidae) would be ideal biological control agents for the Guatemalan potato moth. They have been extensively studied and are usually used as biocontrol agents against lepidopteran pests with successful results. They can also be used along with other control methods, and their mass rearing with alternative hosts is easy [24,25,26]. Furthermore, this genus has been tested and used against other Gelechiids as *Tuta absoluta* (Meyrick, 1917) and *P. operculella*, which are present in potato crops, with promising results [27,28,29].

The efficacy of *Trichogramma* species for biological control of pests depends the suitability of the host target eggs. Therefore, correct election of species/strain has been shown to be essential for a successful release program [30,31]. The appropriate selection of a biological control agent should be based on laboratory, semi-field and field experiments, starting with the testing of indigenous *Trichogramma* species/strains, because they are better adapted to the climate, habitat and host conditions than non-local strains [24,29]. The same approach should be applied under storage conditions. In the present work, one local strain of *T. achaeae* (Nagaraja & Nagarkatti, 1969) and *T. euproctidis* (Girault, 1911) from the Canary Islands were evaluated under laboratory conditions for the first time against *T. solanivora* to evaluate their potential use under field and storage conditions. The present work also represents the first time that acceptance of *T. euproctidis* has been evaluated on *P. operculella*, whereas *T. achaeae* was evaluated in a previous work [26]. Mixed infestations of the two potato moths present in the Canary Islands are common, and both parasitoids may be able to contribute to the control of these pests.

## 2. Materials and Methods

### 2.1. Insects Rearing

Both *Trichogramma* species, *T. achaeae* and *T. euproctidis*, were obtained from local mixed populations from a parasitoid collection program on potato crops carried out by the Canarian Institute of Agricultural Research (Instituto Canario de Investigaciones Agrarias, ICIA).

The mixed population of *T. achaeae* and *T. euproctidis* was separated using using molecular methods (see next section). Purebred rearing of each species was established on *Ephestia kuehniella* eggs at 20–25 ± 1 °C under 60–65% RH with a photoperiod of 16L:8D. Rearing was carried out in Falcon${}^{\mathrm{TM}}$ tubes (v = 50 mL) using modified covers with stainless-steel mesh. Parasitoids were fed a thin line of honey placed on the walls of the tubes.

The two moth species, *T. solanivora* and *P. operculella*, were obtained from local populations on Tenerife Island. During rearing, between 20 and 25 pupae of each species were placed in individual cylindrical plastic glasses ($r_{1}$ = 11.5 cm, $r_{2}$ = 7.5 cm, h = 5 cm; v = 1 L) covered on the upper side with a gauze held by a rubber band and filter paper for oviposition. As a food source, the gauze was moistened with a 50:50 solution of water and honey using a paintbrush. The colonies were kept in climatic chambers at 20 °C under 70% RH in the dark. During the assays, filter papers with the new lays were changed daily to ensure that eggs were less than 24 h old.

Both moths and parasitoids were reared by Agrobiologica S.L. company (Located at the Canarian Institute of Agricultural Research through a collaboration agreement), which periodically provided the insects needed for assays. All insect colonies were started during the year 2021.

### 2.2. Molecular Identification

Two local *Trichogramma* species, *T. achaeae* and *T. euproctidis*, were used in this study. Owing to their small size (<1 mm) and the absence of obvious morphological distinctions between closely related species, morphological identification of Trichogramma is particularly difficult [32]. Therefore, molecular identification was carried out following the approach described by Polaszek et al. (2012) [33]. DNA was extracted from a whole individual specimen using the Chelex protocol [34]. Each specimen was placed in a 1.5 mL Eppendorf tube and ground in 5 µL of proteinase K (10 mg·mL${}^{-1}$) and 75 µL 10% Chelex-100. The tubes were then sealed and incubated overnight at 55 °C. The polymerase chain reaction and sequencing protocols described by Bastin et al. (2021) were followed [35]. The polymerase chain reaction primers for ITS2 are provided by Stouthamer et al. (1999) [36]. Sequences were checked, edited and assembled with CLUSTALW within MEGA software (version 7). The obtained sequences were then compared with the known *Trichogramma* spp. sequences published in the Genbank database: http://blast.ncbi.nlm.nih.gov/Blast.cgi (accessed on 12 March 2023) using BLAST.

At the end of the parasitism period in each bioassay, females were extracted and collected in Eppendorf tubes with alcohol (90%) for subsequent molecular identification using the described method. Every sample of choice and illumination assays were identified, whereas in the rest of the assays, only 5 random samples were identified for each treatment. In all cases, molecular analysis confirmed the correctness the species used in each treatment.

### 2.3. Host Acceptance (No-Choice)

Four “no-choice” (acceptance) bioassays (two with *T. achaeae* and another two with *T. euproctidis* on eggs of *T. solanivora* and *P. operculella*) were carried out following an adapted version of the methodology described by Gallego et al. (2020) [28] (Figure 1). In each trial, 20 newly emerged, mated *Trichogramma* females (less than 24 h old) without previous parasite experience and that had previously been provided with a fine line of honey as a food source were individualized in test tubes (7 cm × 1 cm diameter). In order to ensure their complete sexual maturation and acclimatization, the individualized females were subjected to a 24 h period under assay conditions in a climatic chamber (25 °C, 70% RH, with a photoperiod of 16L:8D).

After the acclimatization period, 30 fresh *T. solanivora* eggs (not sterilized, less than 24 h of age) were glued on white cardboard (0.9 cm × 5.0 cm) in groups of five using a distilled-water-moistened paintbrush. No special glue was used, as the moisture of the paintbrush was enough to keep the eggs fixed. Next, one cardboard section with eggs was introduced to each *Trichogramma* sample. The test tubes were then closed with plastic film and returned to the climatic chamber, where parasitism was allowed for 24 h. At the end of the parasitism period, the females were removed, and development was allowed to continue for seven days in the climatic chamber. Finally, the numbers of parasitized, black (melanized) eggs and surviving larvae were counted. Removal of previously hatched larvae was not necessary because *T. solanivora* do not have a cannibalistic habit.

For statistical analysis, data were checked for homogeneity of variance (F test, Levene test) and the normal distribution of residuals (Shapiro-Wilk test). Next, the average values were compared by ANOVA test using the GLM procedure (Tukey’s test, *p* = 0.05) in IBM${}^{\mathrm{TM}}$ SPSS${}^{\mathrm{TM}}$ Statistics Version 25. In each trial, the experimental design was random and univariate with a single factor at four levels: *T. euproctidis* against *T. solanivora*, *T. euproctidis* against *P. operculella*, *T. achaeae* against *T. solanivora* and *T. achaeae* against *P. operculella*, with 16 replicates.

### 2.4. Host Preference (Choice)

For the choice trials, the same methodology was as for the non-choice trials described above, with one difference: instead of 30 eggs, 15 *T. solanivora* eggs and 15 *P. operculella* eggs localized in opposite corners of the cardboard sections were offered at the same time to each isolated *Trichogramma* female (Figure 1). In these assays, removal of previously hatched larvae was not necessary because neither *T. solanivora* nor *P. operculella* showed cannibalistic or predation relations between them. *Trichogramma achaeae* and *T. euproctidis* host preferences were analyzed using the Manly preference index ($\beta_{2}$), with Chesson (1983) [37] expression:(1)$$\beta_{2}=\frac{\ln\frac{n_{i}-r_{i}}{n_{i}}}{\sum_{2}^{1}\ln\frac{n_{i}-r_{i}}{n_{i}}}$$
where $\beta_{2}$ is the Manly preference index (without replacement), $r_{i}$ is the number of parasitized eggs, $n_{i}$ is the number of exposed eggs and i=1 or 2 represents the host eggs of each tested species. A preference index value of $\beta_{2}$ > 0.5 corresponds to preference, $\beta_{2}$ = 0.5 indicates indifference and $\beta_{2}$ < 0.5 represents rejection.

The number of parasitized eggs was analyzed by ANOVA test using the GLM procedure, and the average values were compared by Tukey’s test (*p* = 0.05) and Manly preference index with one-sample *t*-test against a hypothetical mean of 0.5. Previously, the homogeneity of variance (F test and Levene test) and normal distribution of residuals (Shapiro–Wilk test) were checked. These analyses were carried out with IBM${}^{\mathrm{TM}}$ SPSS${}^{\mathrm{TM}}$ Statistics Version 25.

### 2.5. Functional Response and Parasitism

The functional responses of *T. achaeae* and *T. euproctidis* were determined at 15, 20, 25 and 27 ± 1 °C at densities of 10, 30, 50, 70 and 90 *T. solanivora* eggs under 70% RH with a photoperiod of 16L:8D. For theses assays, the same procedures explained in the previous section were followed, but the data were collected upon the emergence of the *Trichogramma* adult. The following data were registered: parasitized eggs, emerged host larvae, collapsed eggs and number of emerged *Trichogramma* males and females. A total 10–15 replications were performed for each egg density in each *Trichogramma* species.

The data of parasitized eggs concerning to the functional response of both *Trichogramma* species at each tested temperature were subjected to two different statistical analyses. First, the the homogeneity of variance and normality of the data were checked with the same procedure as in the previous assays; then, the density factor significance was analyzed by GLM analysis, and the mean values were compared using Tukey’s test (*p* = 0.05) in IBM${}^{\mathrm{TM}}$ SPSS${}^{\mathrm{TM}}$ Statistics Version 25. Next, equations of functional response types I, II and III were adjusted for *T. achaeae* and *T. euproctidis* at all tested temperatures according to the following expressions [38,39,40]:Type I:
(2)$$N_{a}=N_{t}\left[1-\exp\left(-a^{\prime }\cdot T\cdot P_{t}\right)\right]$$Type II:
(3)$$N_{a}=N_{t}\cdot \left(1-\exp\left[-\frac{a^{\prime }\cdot T\cdot P_{t}}{1+a^{\prime }\cdot T_{h}\cdot N_{t}}\right]\right)$$Type III:
(4)$$N_{a}=N_{t}\cdot \left(1-\exp\left[-\frac{\alpha \cdot T\cdot N_{t}\cdot P_{t}}{1+\alpha \cdot T_{h}\cdot N_{t}+\alpha \cdot T_{h}\cdot N_{t}^{2}}\right]\right)$$
where $N_{a}$ is the number of parasitized hosts, $N_{t}$ is the density of the host or prey, $a^{\prime }$ is the instantaneous search rate (equivalent to Nicholson-Bailey’s “area of discovery”: $a=a^{\prime }T$, $days^{-1}$; *T*) on day-1, *T* is the total available search time (days), $P_{t}$ is the number of parasitoids, $T_{h}$ is the host manipulation time and α is parasitoid mortality potential. The cited adjustments were performed with Tablecurve 2D software, version 5.0.

The corrected Akaike information criterion ($AIC_{c}$) was applied to select the best adjustment model because it supposes better statistical precision for comparisons between models than the regression coefficient ($r^{2}$) [41]. However, the latter was calculated to determine the goodness of each performed adjustment.

For parasitism and the sex ratio, the differences at the assayed temperatures (15, 20, 25 and 27 °C) in each species and between species, the number of parasitized eggs and the percentage of females were analyzed by two-way ANOVA and Tukey’s test (*p* = 0.05) with the GML procedure.

### 2.6. Illumination Assays

Two illumination acceptance assays were conducted under light conditions, and two assays were conducted under dark conditions for *T. achaeae* and *T. euproctidis* following the same methodology as that applied in the no-choice assay, with the exception of the photoperiod, which was 24L:0D under light conditions and 0L:24D in dark assays.

In each trial, the experimental design was random and univariate with a single factor at two levels: parasitized eggs under light condition versus parasitized eggs under darkness conditions, with 20 replicates. Because the Shapiro-Wilk test confirmed the non-normality of the data, the numbers of parasitized *T. solanivora* eggs under illuminated and darkness conditions were compared using a non-parametric Kruskal–Wallis one-way analysis with a level of significance of *p* = 0.05. Both analyses were carried out with IBM${}^{\mathrm{TM}}$ SPSS${}^{\mathrm{TM}}$ version 25.

### 2.7. Search Ability under Semi-Field Conditions

The search ability assay (Figure 2) was carried out under greenhouse conditions in insect cages (45 × 45 × 45 cm). Twenty potatoes (washed and numbered) were placed in 15 L trays and covered with a 5 cm layer of a mixed substrate of orchard soil and lapilli (2:1), previously disinfected with a steam machine (90 °C, 45 min). Three treatments were carried out (control, *T. achaeae* and *T. euproctidis*) with four replicates per treatment. Five couples (male and female) of *T. solanivora* were released in all treatments. The first release of 60 females of *T. achaeae* and 60 of *T. euproctidis* was carried out after a period of 48 h in each treatment. To obtain *Trichogramma* females, *T. solanivora* eggs developed for eight days at 25 °C were parasitized based on the sex ratio, as indicated in the “Results” section. This procedure yielded 80 and 76 parasitized eggs for *T. achaeae* and *T. euproctidis*, respectively. The second *Trichogramma* release was performed seven days later, with the same number of females of each species. No wasps were released in the control treatment. After a parasitism period of 20 days, the tubers were transferred to plastic boxes for 25 extra days to allow for the emergence of most of the larvae. The mean temperature during the greenhouse assay was 23.18 ± 5.30 °C, whereas the RH was 67.38 ± 16.06%.The evaluation of the assay consisted of counting, damaged tubers, number of mines per tuber and the survival of *T. solanivora* individuals (larvae, pupae and adults). Additionally, tubers were cut to search for any larvae remaining inside, which were also counted.

For statistical analyses, once the variance homogeneity and normality of data (Levene test and Shapiro-Wilk test, respectively) were checked, the mean number of survivors were compared by ANOVA test using the GLM procedure (Tukey’s test, *p* = 0.05). In the case of the number of mines per potato and undamaged tubers, the data were analyzed by a generalized linear model (GZLM) using the Poisson distribution function and the log linear link function.

## 3. Results

### 3.1. Host Acceptance (No-Choice)

Statistical analysis showed the major parasitism capacity of *T. euproctidis* and *T. achaeae* against *P. operculella* relative to that of *T. solanivora*, with a higher number of parasitized *T. solanivora* eggs with *T. euproctidis* than with *T. achaeae* (F = 50.72; df = 3, 59; *p* < 0.05) under no-choice conditions.

Figure 3 shows the parasitism level of *T. euproctidis* and *T. achaeae* against *T. solanivora* and *P. operculella* within 24 h under no-choice conditions.

### 3.2. Host Preference (Choice)

Significant differences were found in mean values of the number of parasitized eggs in both species. *Trichogramma euproctidis* showed a higher number of parasitized eggs for *T. solanivora* than for *P. operculella*: 10.87 ± 0.58 vs. 4.81 ± 0.56 (F = 52.03; df = 1, 29; *p* < 0.05). The same pattern was observed for *T. achaeae*, with 6.35 ± 0.33 parasitized *T. solanivora* eggs vs. 3.88 ± 0.74 for *P. operculella* (F = 5.27; df = 1, 32; *p* < 0.05). The Manly preference index (MI) showed the preference of *T. euproctidis* and *T. achaeae* for *T. solanivora* against *P. operculella* (t = 4.65; *p* = 0.001 and t = 2.03; *p* = 0.001, respectively).

Figure 4 shows the parasitism level of *T. euproctidis* and *T. achaeae* against *T. solanivora* and *P. operculella* eggs exposed for 24 h under choice conditions.

### 3.3. Functional Response

#### 3.3.1. *Trichogramma euproctidis*

Analysis of functional response showed that at all the tested temperatures (15, 20, 25 and 27 °C), the parasitic behavior of adult females of *T. euproctidis* corresponded with type III. The adjustments showed the lowest values in the corrected Akaike indices ($AIC_{c}$). With respect to the adjustment parameters, the handling times ($T_{h}$) were similar; on the other hand, the mortality potential ($\alpha^{\prime }$) showed similar values at the three highest temperatures (Table 1). The graphs in Figure 5 do not look like the typical “S” curves of functional response type III. It’s because that the inflexion point of the type III curves, from which the parameter “α” or parasitoid mortality potential occurs at very low densities of host eggs offered in the species *T. solanivora*.

Analysis of variance showed a significant effect of the host density on parasitism for all tested temperatures (F = 26.47; df = 4, 5; *p* < 0.05 at 15 °C; F = 25.40; df = 4, 42; *p* < 0.05 at 20 °C; F = 31.78; df = 4, 50; *p* < 0.05 at 25 °C and F = 33.71; df = 4, 45; *p* < 0.05 at 27 °C, respectively). At every temperature except 27 °C, *T. euproctidis* reached the maximum parasitism activity with 30 eggs.

Figure 5 shows the functional response curves of *T. euproctidis* against *T. solanivora* at four temperatures (15, 20, 25 and 27 °C).

#### 3.3.2. *Trichogramma achaeae*

*Trichogramma achaeae* showed a type II functional response at the three lowest temperatures (15, 20 and 25 °C) that changed to type III at the highest temperature (27 °C). Thus, in relation to the adjustment parameters, in the present case, only the manipulation times ($T_{h}$) can be compared. As shown in Table 2, the values decreased with increased temperature. In Figure 2, the graph at 27 °C, the typical sigmoidal shape of type III functional response cannot be seen for the same reason explained for *T. euproctidis* in the previous section.

With regard to host density, a significant influence on the number of eggs was observed at the four evaluated temperatures F = 5.22; df = 4, 46; *p* < 0.05 at 15 °C; F = 11.70; df = 4, 46; *p* < 0.05 at 20 °C; F = 15.83; df = 4, 46; *p* < 0.05 at 25 °C and F = 9.65; df = 4, 49; *p* < 0.05 at 27 °C). *T. achaeae* reached the maximum parasitism level at all tested temperatures at a density of 30 eggs.

Figure 6 shows the functional response curves of *T. achaeae* against *T. solanivora* at the four tested temperatures (15, 20, 25 and 27 °C).

### 3.4. Parasitism

Analysis of the number of parasitized eggs (Figure 7) shows a significant effect of the species and the temperature (F = 11.75; df = 1, 74; *p* < 0.05 and F = 11.75; df = 3, 74; *p* < 0.05, respectively), with no effect of the interaction. *T. euproctidis* showed the highest parasitic activity between 20–27 °C, with significant differences relative to 15 °C (F = 4.69; df = 3, 37; *p* < 0.05). In the case of *T. achaeae*, the highest level of parasitic activity was found at 20 and 25 °C (F = 6.37; df = 3, 36; *p* > 0.05). These temperatures only presented significant differences relative to 15 °C (F = 6.37; df = 3, 36; *p* < 0.05). On the other hand, analysis of the number of eggs parasitized by *T. euproctidis* compared with *T. achaeae* at each temperature showed higher mean values for *T. euproctidis* at all tested temperatures: 15, 20, 25 and 27 °C (F = 38.07; df = 1, 20; *p* < 0.05, F = 13.73; df = 1, 18; *p* < 0.05, F = 6.37; df = 1, 18; *p* < 0.05 and F = 45.14; df = 1, 17; *p* < 0.05, respectively).

### 3.5. Sex Ratio and Fecundity

Statistical analysis of both *Trichogramma* progeny showed a significant influence of temperature and species (F = 5.57; df = 3, 271; *p* < 0.05 and F = 5.57; df = 1, 271; *p* < 0.05, respectively) with no effect of the interaction interaction (F = 5.57; df = 3, 271; *p* > 0.05) on the sex ratio. In *T. euproctidis* didn’t find significant differences with temperature (F = 1.29; df = 3, 137; *p* > 0.05). In contrast, *T. achaeae* showed significant differences at 27 °C compared to 15 and 20 °C (F = 8.08; df = 3, 134; *p* < 0.05). However, a comparison of the two *Trichogramma* species only showed significant differences at the lower assayed temperatures (15 and 20 °C) with a higher number of *T. euproctidis* females (F = 6.37; df = 1, 63; *p* < 0.05 and F = 12.27; df = 1, 63; *p* < 0.05, respectively) (Figure 8).

Fecundity results show the emergence of between one and two adults by each parasitized egg for both species at all the assayed temperatures (Table 3).

Table 3 shows the number of emerged adults at each of the tested temperatures in functional response assays for *T. euproctidis* and *T. achaeae*.

**Table 3 insects-14-00679-t003:** Mean number (±SE) of emerged adults per parasitized egg obtained in the functional response assays at each temperature for *T. euproctidis* and *T. achaeae*.

Species	15 °C	20 °C	25 °C	27 °C
*T. euproctidis*	1.08 ± 0.3	1.20 ± 0.04	1.26 ± 0.04	1.27 ± 0.04
*T. achaeae*	1.41 ± 0.11	1.16 ± 0.04	1.13 ± 0.05	1.23 ± 0.04

Figure 8 shows the percentage of female *T. euproctidis* and *T. achaeae* at the tested temperatures in functional response assays.

**Figure 8 insects-14-00679-f008:**
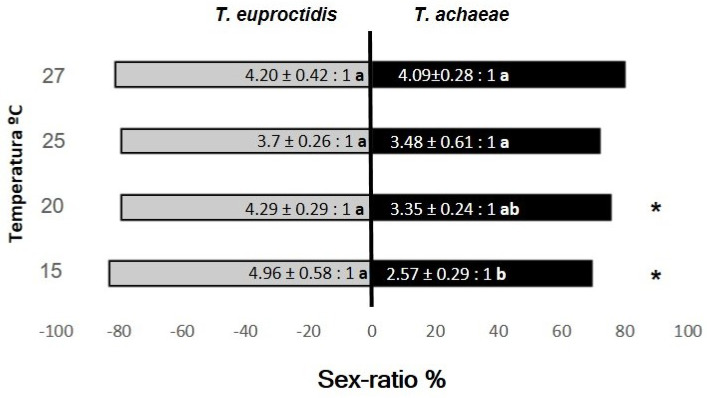
Mean of sex ratio (F:M) (±SE) obtained for *T. achaeae* and *T. euproctidis* with temperature over *T. solanivora* eggs. Different lowercase letters indicate significant differences between temperatures (two-way ANOVA and Tukey’s test, *p* = 0.05) in each species. “*” indicates significant differences between the two species at the same temperature (*t*-test, *p* = 0.05).

### 3.6. Illumination Acceptance Assays

The parasitism activity analysis result did not show significant differences between light and darkness conditions for *T. euproctidis* (Kruskal–Wallis test, $X_{2}$ = 125.00, *p* > 0.05). In contrast, *T. achaeae* was unable to parasitize the host eggs under darkness conditions (Kruskal–Wallis test, $X_{2}$ = 210.00, *p* < 0.05) (Figure 9).

Figure 9 shows the average number of parasitized *T. solanivora* eggs for *T. euproctidis* and *T. achaeae* under light and darkness conditions.

### 3.7. Search Ability under Semi-Field Conditions

The statistical analysis results show significant differences in all the evaluated parameters between *T. achaeae* and control treatments: *T. solanivora* survival (F = 6.25; df = 2, 9; *p* < 0.05, Figure 10), number of mines per tuber (omnibus likelihood ratio $X_{2}$ = 7.459; df = 2; *p* = 0.24) and percentage of undamaged tubers (omnibus likelihood ratio $X_{2}$ =14.004; df = 2; *p* = 0.001) (Figure 11). *Trichogramma euproctidis* did not differ significantly in any parameter compared to the control (omnibus likelihood ratio $X_{2}$ = 7.459; df = 2; *p* = 0.62 for number of mines; omnibus likelihood ratio $X_{2}$ = 14.004; df = 2; *p* = 0.24 for undamaged tubers; and F = 6.25; df = 2, 9; *p* = 0.137, for the number of survivors) and presented significant differences with respect to *T. achaeae* in the number of mines per tuber (omnibus likelihood ratio $X_{2}$ = 14.004; df = 2; *p* = 0.04) but not in the percentage of undamaged tubers (omnibus likelihood ratio $X_{2}$ = 14.004; df = 2; *p* = 0.056) (Figure 10 and Figure 11).

Figure 10 shows the mean number of surviving *T. solanivora* obtained in the control, *T. euproctidis* and *T. achaeae* treatments under semi-field conditions.

**Figure 10 insects-14-00679-f010:**
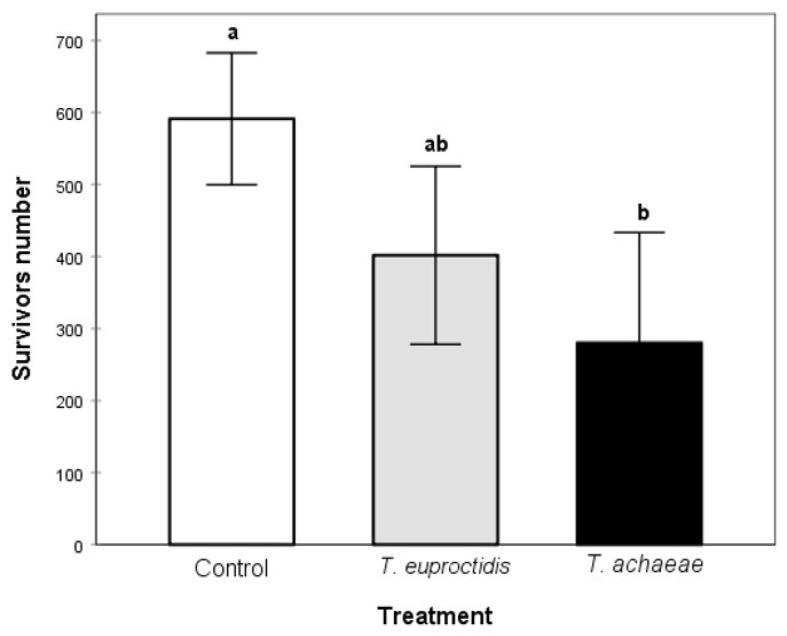
Mean (±SE) values of the number of surviving *T. solanivora*. Values with different letters indicated significant differences (one-way ANOVA and Tukey’s test, *p* = 0.05).

Figure 11 shows the damages observed in the control, *T. euproctidis* and *T. achaeae* treatments, as well as the mean number of mines per tuber and the percentage of undamaged tubers.

**Figure 11 insects-14-00679-f011:**
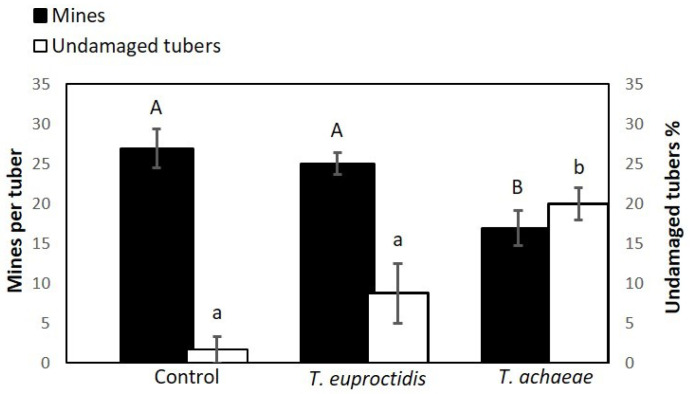
Mean values of the number of mines per tuber (±SE) and the percentage of undamaged tubers (±SE) obtained in each treatment. Different letters indicate significant differences (omnibus test, *p* = 0.05).

## 4. Discussion

In the present work, with the main objective of determining the potential application of *T. achaeae* and *T. euproctidis* as biological control agents against the Guatemalan potato moth (*T. solanivora*) in field and storage conditions, an evaluation and selection procedure were carried out. The applied methodology analyzed the four steeps defined for Vinson (1976) [42] in host selection process of parasitoids: host habitat location, host location, host acceptance and host suitability. For the first steps, i.e., habitat location and host location, because the analyzed *Trichogramma* species were recollected on potato crops, it is reasonable to assume that the appropriate habitat corresponds to the field conditions rather than artificial storage conditions.

Host acceptance was analyzed using the non-choose assays. As mentioned in the Introduction, mixed infestations of *T. solanivora* and *P. operculella* are common. In such contexts, both host are available for parasitic activity of the analyzed *Trichogramma* spp. The presence of an alternative compatible host is known to result in a reduction in parasitism efficacy against the target pest and undesired ecological effects. Therefore, achieve maximum success in biological control programs, the selection of a candidate with a strong preference for the target pest is important [24,43,44]. In the present work, we confirmed that *T. achaeae* and *T. euproctidis* have good acceptance for both hosts (Figure 2). This is the first report of *T. solanivora* as a compatible host for the studied parasitoids, i.e., *T. achaeae* and *T. euproctidis*. Our study also confirmed the acceptance of *P. operculella* by *T. achaeae*, which was previously reported by Gallego et al. (2019) [28]. In all cases, both parasitoids showed significantly higher parasitism levels against *P. operculella* under no-choice conditions, and the major efficacy of *T. euproctidis* compared to *T. achaeae* was observed against both hosts. When the tested *Trichogramma* species were exposed to eggs of the two moths at the same time, both parasitic wasps showed a considerable number of parasitized eggs over Guatemalan potato moth, in agreement with the results of the Manly index, which indicated a clear preference for this host (Figure 4). The obtained preference results are interesting because under field conditions *P. operculella*, eggs are present on the leaves and more accessible than *T. solanivora* eggs, which are laid in the soil [3]. The major preference for Guatemalan potato moth eggs could facilitate wasps to find them in mixed infestations. Multiple factors have been pointed to with respect to the egg preference of *Trichogramma* species, such as egg age, chorion thickness and hardness, among other factors. However, previous works found that when *Trichogramma* are exposed to different host eggs, they usually select the largest eggs, most likely because a larger host size is preferred because parasitoids recognize that there are more nutrients available for their progeny [31,43,45,46]. This phenomenon is consistent with our results, i.e., that *T. solanivora* eggs are larger than *P. operculella* eggs (around 0.41 × 0.53 mm, compared to 0.50 × 0.35 mm [45,47,48]). In the evaluation of acceptance and preference, some authors have also counted the number of wasp-egg contacts; however, their results showing correspondence between high levels of parasitism and a high number of contacts were inconsistent [27,30,45]. Therefore, wasp-egg contacts does not seem to be a clear criterion.

The last step, i.e., host suitability, is concerned with biological factors related to the development of the parasitoid in the potential host, i.e., survival, parasitism rate, sex-ratio of progeny and emergence, which are strongly influenced by temperature [45,49]. Another factor in the selection of a pest control agent is the functional response, i.e., the number of eggs parasitized in response to host densities [50,51], which is a basic factor that allows contributes to the understanding of host interactions and is among the main influences on the stability of the system [52]. Functional response can be categorized into three forms: type I, linear increase in killed host/prey to a plateau; type II, a curvilinear rise to a plateau that then levels off under the influence of handling time or satiation; and type III, a sigmoidal increase in attacked hosts [53,54,55]. In our study, the parasitoids showed different patterns in functional response, whereas *T. euproctidis* maintained type III (Table 1), and *T. achaeae* presented type II at the three lower temperatures and changed to type III at 27 °C (Table 2). The type II functional response, as shown by *T. achaeae*, is the most common response found in trichogrammatidae and has been suggested to be associated with suitable parasitoid hosts [56,57,58]. This response result is in accordance to those reported by several authors at a temperature of 25 °C [50,59,60]. However, our results contrast with the type I response reported in previous work for *T. achaeae* over *Ephestia kuehniella*, likely reflecting the effect of the different host [50]. In the same work, the author showed a change in functional response to type II at 25 °C, becoming type II to III at 27 °C, as reported by Wang and Ferro (1998) [61] in *Trichogramma ostridina*. This case illustrates the influence of temperature on functional response. The constant type III response at all temperatures shown by *T. euproctidis* is a reflection of a major potential against *T. achaeae* because type III is considered the best kind of behavior to obtain a successful biological control of pests because it is theoretically the only response that can achieve direct density dependence at a low population host level and stabilize the host-parasitoid system [57]. The potential of *T. euproctidis* in functional response is in accordance with parasitic activity, as this wasp showed a higher parasitism level than *T. achaeae* at all temperatures (Figure 7). A possible justification for the parasitization behavior of *T. euproctidis* in relation to *T. achaeae*, under laboratory conditions is the different manipulation times ($T_{h}$) (the sum of drumming, drilling and oviposition), as found in the functional response (Table 1 and Table 2); thus, the first species has a host manipulation time that ranges from 0.0372 to 0.0418 days compared to second species, with values ranging from 0.0540 to 0.0845 days, meaning that *T. euproctidis* spends less manipulation time saving within the day, leaving more time for searching and finding more host eggs. Handling time was previously identified as an important factor in parasitization by *Trichogramma* species [62].

We observed that both parasitoids reach the maximum parasitism level at the density of 30 eggs, except *T. euproctidis* at 25 °C, which was at the density of 50 eggs, but only parasitized 21 eggs. Using different combinations of *Trichogramma* species and hosts, other authors also found that parasitoids could not parasitize all eggs within 24 h when 30 host eggs were exposed [49,50,59,60], meaning that 30 eggs is the optimal offer for comparison of *Trichogramma* species.

Our results only reflect an influence of temperature on the sex ratio in *T. acaheae* at 27 °C, showing a higher number of females, whereas previous works obtained different conclusions about the effect of temperature on the number of female progeny [49,50,63,64], probably indicating a greater influence of parasitoid and host species than temperature. Comparing both species, we found a higher number of females on *T. euproctidis* compared to *T. achaeae* at 15 and 20 °C (Figure 8). This is an interesting result because it is the usual range of small growers’ warehouses, who usually store their harvest at room temperature. Under all evaluated temperature conditions, the parasitoids clearly exceeded the minimum of 50% female progeny recommended by the IOBC [65].

Regarding fecundity, our results did not show any influence of temperature. Between one and two emerged adults per parasitized egg was observed under all tested conditions for both species, indicating a high level of host suitability, over the 80% recommended by the IOBC, although considered a minor factor in inundative control strategies, which is the case for *Trichogramma* species [27,65].

In spite of the better values shown by *T. euproctidis* for most of the analyzed biological parameters in the laboratory evaluations, in the semi-field assay, this species did not reduce *T. solanivora* populations or prevent damage in tubers. However, *T. achaeae* reduced the pest population, reaching an efficacy around 50%, and yielding 20% of undamaged tubers (Figure 9). Taking into consideration the fact that the same number of individuals of both species was released and that the parasitic potential was higher at all tested temperatures for *T. euproctidis* under laboratory conditions, these results suggest a lower searching ability in soil of this wasp under field conditions. Therefore, even when it can access the host, it is not an effective pest control agent. Although the importance of kairomones in host search and selection and their role in biological pest control are not mentioned in this paper, they have been addressed by other authors [66]. In this regard, the superior effectiveness of *T. achaeae* in relation to *T. euproctidis* in the semi-field test may be due to the fact that the first species presents a greater attraction, as well as a greater stimulus in response to oviposition by the kairomones of the host species than the second species, as has been reported for other species and habitat conditions for this group of egg parasitoids [67]. In contrast to semi-field conditions, the illumination assays showed the inability of *T. achaeae* to parasitize under darkness conditions compared to *T. euproctidis*, which kept its potential unchanged under both light and darkness conditions. These results indicate that *T. achaeae* is not suitable for storage conditions.

## 5. Conclusions

In the present study, a systematic methodology was applied to select and evaluate two *Trichogramma* species (*T. achaeae* and *T. euproctidis*) against the Guatemalan potato moth (*T. solanivora*).

In the laboratory assays (host acceptance, host preference and functional response), *T. euproctidis* showed better performance than *T. achaeae* in controlling the pest. However, in semi-field trials, *T. euproctidis* did not reduce *T. solanivora* populations or protect tubers from the pest, whereas *T. achaeae* behaved as an effective biocontrol agent under field conditions, suggesting a higher host searching ability or attraction to the host than that of *T. euproctidis*. Nonetheless, only *T. euproctidis* showed parasitic activity in the dark. Therefore, this species will be evaluated under semi-storage conditions in future studies.

The results obtained in this work highlight the relevance of performing assays with natural enemies under semi-field/semi-storage conditions before testing them in the field or in storage facilities. The selection of pest biocontrol agents based only on positive laboratory assay results could lead to incorrect decisions and unnecessary expense of effort and resources.

## Figures and Tables

**Figure 1 insects-14-00679-f001:**
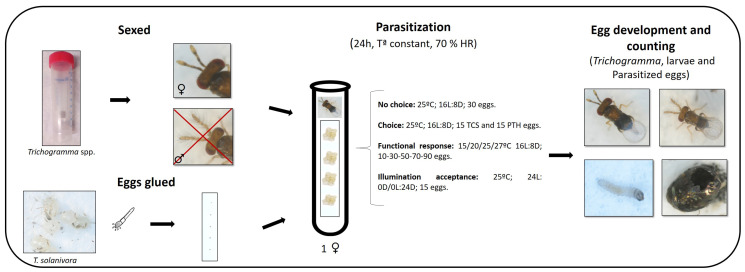
Summary of laboratory assay methods (no-choice, choice, functional response and illumination acceptance). TCS = *T. solanivora*; PTH = *P. operculella*.

**Figure 2 insects-14-00679-f002:**
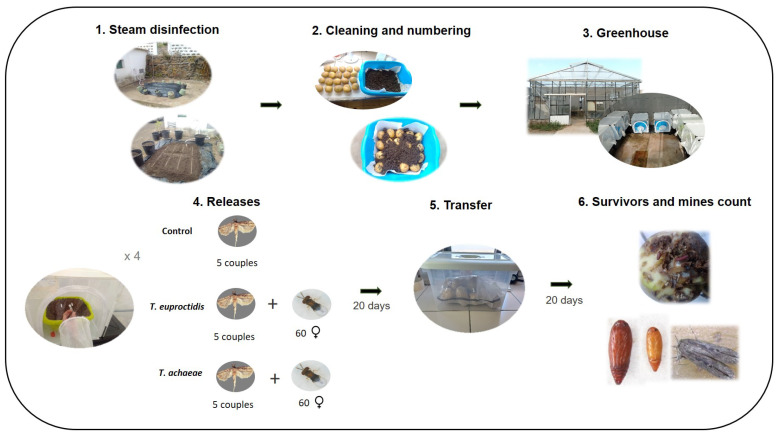
Summary of the searching semi-field assay.

**Figure 3 insects-14-00679-f003:**
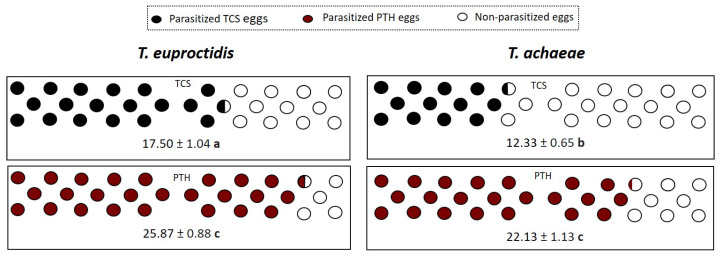
Mean number of parasitized *T. solanivora* (TCS) and *P. operculella* (PTH) eggs (±SE) when which are exposed to *T. euproctidis* and *T. achaeae* females at 25 °C under 70% RH with 16L:8D. Different letters denote significant differences in the number of parasitized eggs between treatments with different combinations of parasitoids (*T. euproctidis* or *T. achaeae*) and egg hosts (*p* = 0.05).

**Figure 4 insects-14-00679-f004:**
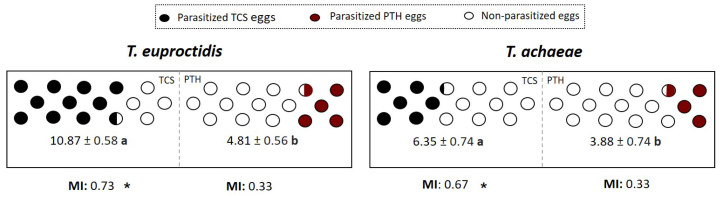
Number of parasitized *T. solanivora* (TCS) and *P. operculella* (PTH) eggs (±SE) and obtained Manly Index (MI) when which are exposed to *T. euproctidis* and *T. achaeae* females the same time at 25 °C under 70% RH with 16L:8D. Different lowercase letters represent significant differences in the number of parasitized eggs (one-way ANOVA and Tukey’s test; *p* = 0.05), and “*” indicates significant preference (*t*-test; *p* < 0.05) of each parasitic wasp.

**Figure 5 insects-14-00679-f005:**
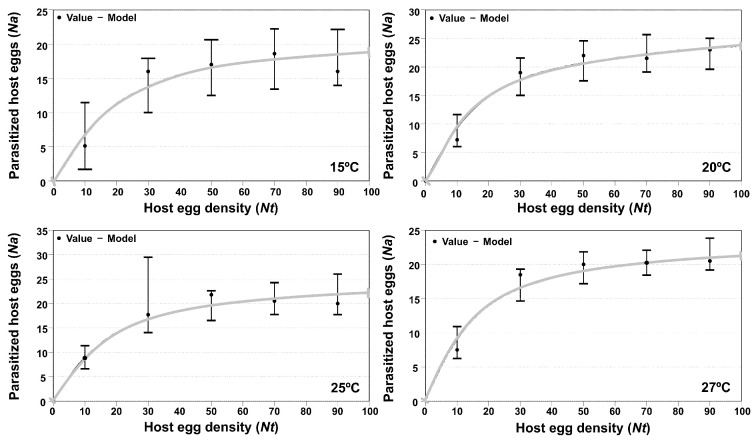
Functional response of *T. euproctidis* female parasitizing *T. solanivora* eggs at four temperature levels (15 °C, 20 °C, 25 °C and 27 °C) under laboratory conditions (70% RH and 16L:8D). Vertical bars indicate 95% confidence intervals. Note: For a correct interpretation of the figures it should be taken into account that the *y*-axis scales may be different.

**Figure 6 insects-14-00679-f006:**
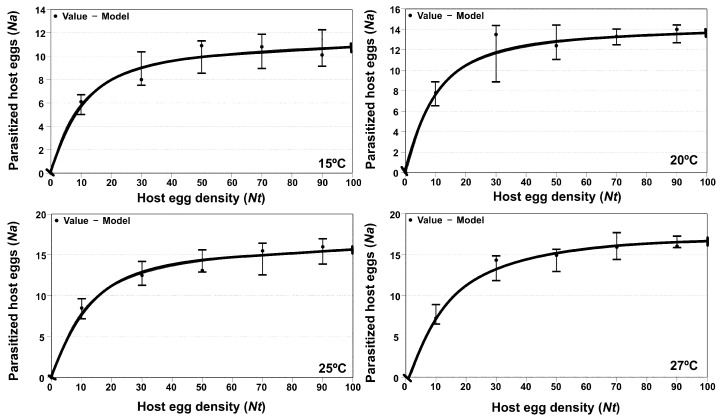
Functional response of *T. achaeae* female parasitizing *T. solanivora* eggs at four temperature levels (15 °C, 20 °C, 25 °C and 27 °C) under laboratory conditions (70% RH and 16L:8D). Vertical bars indicate 95% confidence intervals. Note: For a correct interpretation of the figures, it should be taken into account that the *y*-axis scales may be different.

**Figure 7 insects-14-00679-f007:**
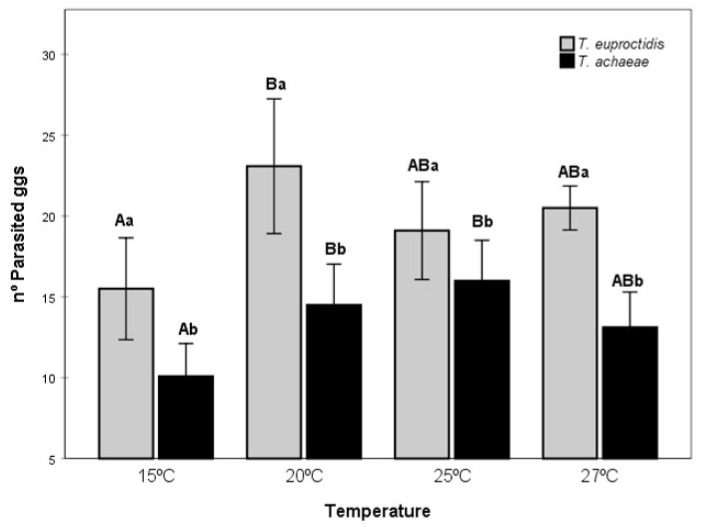
Mean number of eggs parasitized by *T. solanivora* (±SE) for *T. euproctidis* compared to *T. achaeae* at a host density of 90 eggs in the temperature range of 15–27 °C. Different uppercase letters represent differences in the number of parasitized eggs among the tested temperatures in each species. Different lowercase letters indicate significant differences between the two species (two-way ANOVA and Tukey’s test, *p* = 0.05).

**Figure 9 insects-14-00679-f009:**
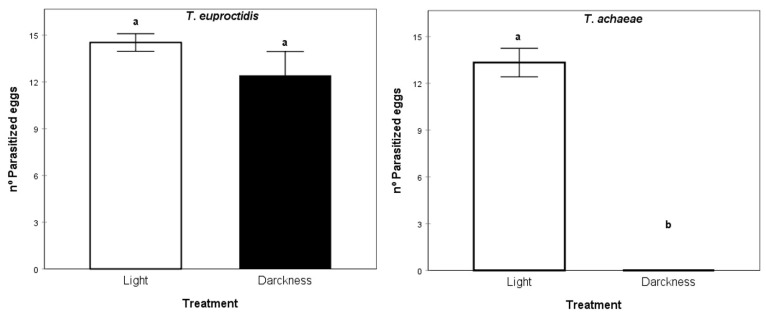
Mean number of *T. solanivora* eggs parasitized (±SE) by *T. euproctidis* and *T. achaeae* under light conditions compared with darkness conditions at 25 °C and 70% RH. Values with different letters indicated significant differences (Kruskal-Wallis test, *p* = 0.05).

**Table 1 insects-14-00679-t001:** Parameters and statistical significance for the type I, II and III functional response equations when different densities of *T. solanivora* eggs were exposed to the adult female of *T. euproctids* for 24 h under laboratory conditions. “*” indicates the lowest $AIC_{c}$ value at each temperature.

Temperature	Type	Fit Curve Parameters			Statistical Parameters		
		$a^{\prime }$	α	$T_{h}$	d.f.	$R^{2}$	${\mathrm{AIC}}_{C}$
	I	0.5155	-	-	5	0.8052	7.30899
**15 °C**	II	5.9158	-	0.0466	4	0.8845	8.77405
	III	-	0.2569	0.0491	4	0.9053	6.96387 *****
	I	0.7024	-	-	5	0.8681	7.33837
**20 °C**	II	1750.9	-	0.0372	4	0.9647	6.52153
	III	-	$3.8\times 10^{16}$	0.0372	4	0.9653	−0.6938 *****
	I	0.6725	-	-	5	0.8232	7.78757
**25 °C**	II	4847.2	-	0.0436	4	0.9302	7.90383
	III	-	$7.78\times 10^{11}$	0.0394	4	0.9557	6.52805 *****
	I	0.6218	-	-	5	0.7786	8.29285
**27 °C**	II	$2.27\times 10^{15}$	-	0.0412	4	0.9655	6.04313
	III	-	$9.07\times 10^{13}$	0.0418	4	0.9661	5.76405 *

**Table 2 insects-14-00679-t002:** Parameters and statistical significance for the type I, II and III functional response equations when different densities of *T. solanivora* eggs were exposed to the adult female of *T. achaeae* for 24 h under laboratory conditions. “*” indicates the lowest $AIC_{c}$ value at each temperature.

Temperature	Type	Fit Curve Parameters			Statistical Parameters		
		$a^{\prime }$	α	$T_{h}$	d.f.	$R^{2}$	${\mathrm{AIC}}_{C}$
	I	0.2767	-	-	5	0.6347	5.49962
**15 °C**	II	3.4655	-	0.0845	4	0.9543	2.30479 *
	III	-	0.6533	0.0876	4	0.9515	2.48573
	I	0.3882	-	-	5	0.3079	9.34299
**20 °C**	II	$2.23\times 10^{8}$	-	0.0680	4	0.9660	2.93269 *
	III	-	$1.48\times 10^{19}$	0.0651	4	0.9608	2.99312
	I	0.3835	-	-	5	0.3843	8.68488
**25 °C**	II	19.4182	-	0.0585	4	0.9782	2.22521 *
	III	-	$2.55\times 10^{17}$	0.0588	4	0.9775	2.26747
	I	0.4465	-	-	5	0.6211	8.22257
**27 °C**	II	26.851	-	0.0547	4	0.9766	2.96402
	III	-	1.0972	0.0540	4	0.9783	2.73542 *

## Data Availability

The data presented in this study are available on demand from the first author at jorgegavara@gmail.com.

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
