# Peer review of "Evaluation and Selection of New Trichogramma spp. as Biological Control Agents of the Guatemalan Potato Moth (Tecia solanivora) in Europe"

_insects, 2023, doi:10.3390/insects14080679_

Round 1

Reviewer 1 Report

Overall, I found this paper to be a significant contribution to the biological control community. I have no major comments other than for Figure 10. It looks like T. euproctidis should be labeled as 'ab' since it is not significantly different from the control and T. achaeae.

The major drawback of this study, and is one that must be remedied before the paper is accepted, is the grammar. The paper is difficult to read/understand. The paper needs to be gone through by a native English speaker to correct the language. This factor affected my ability to give a comprehensive review. I initially tried to clean up the grammar, but it became too much of a task as the paper went on. Regardless, I think this is a valuable study that should be published, but only after the grammatical problems are resolved.

The paper needs to undergo extensive editing before acceptance. There are numerous grammatical errors throughout. The manuscript needs to be thoroughly vetted by a native English speaker.

Reviewer 2 Report

Vidal et al. test two Trichogramma species against Tecia solanivora and compare wasp responses to T. solanivora and P. operculella eggs. They conclude that T. achaeae could prove useful under field conditions whereas T. euproctidis may be useful in potato storage facilities. This study will be of interest to readers of Insects.
I have several issues that need to be addressed before the manuscript is suitable for publication.
1.    The text contains many grammatical errors and typos. It is also excessively wordy in many parts. I would ask the authors to synthesize their ideas and present them concisely.
2.    The most common name for T. solanivora in English is the "Guatemalan POTATO moth", although other names are also used.
3.    L76. should be able to deal with them? You mean ....may be able to contribute to the control of these pests?
4.    Section 2.1 When were the wasp colonies started?
5.    L94. Colonies, not breeds.
6.    Section 2.2. Please explain why molecular identification was required. Are these species morphologically difficult to distinguish apart?
7.    L133. Indicate type of glue.
8.    Use of ANOVAs – please explain how you ensured that data met the assumptions of ANOVA or how the suitability of GLMs was checked (applies throughout the manuscript).
9.    Section 2.7. Reword to 'Search ability' or 'Search capacity under semi-field conditions'.
10.    Fig 3, Fig 4. It took me a minute to understand this figure as the use of circles (solid or open) is not obvious. I would ask the authors to present these results as standard bar graphs for clarity.
11.    F statistics. F statistics can only be understood when presented together with the treatment and ERROR degrees of freedom. Please modify this throughout the manuscript.
12.    Figure 5. Vertical bars indicate 95% confidence intervals (not p = 0.05).
13.    Table 1, Table 2. What do vales in bold indicate?
14.    L275, Table 1. At 27 degrees, the type II response has a lower AIC value, so why did you state that the wasp showed a type III response?
15.    Figures 7, 9, 10, use commas not decimal points on the y-axis. Please change to points.
16.    L301, Table 3. I do not agree with this statement. Values in Table 3 are 1.08 – 1.41 indicating that a fraction of the eggs produced two parasitoids in all cases.
17.    Discussion. L332 – L354 This is a very long preamble (22 lines of text) before the authors even begin to discuss their results. The Discussion should begin with mentioning the main findings.
18.    L370-71. Why do parasitoids usually select larger hosts? The reasons are well understood. This also has implications for sex allocation, doesn't it?
19.    L378-412. This paragraph fails to address the major issue (for me) - - why do functional responses appear to be type II in all cases, but the author's analysis assigns them to type III class responses. What are the processes that could be generating reduced parasitism at low host densities and why don't we see this in the plots of Figs 5 and 6? Is there a significant difference between type II and type III models in all cases - - I see only marginal improvements in AIC values. The type II model is clearly the more parsimonious model. I saw no evidence for type III responses in the empirical data.
20.    The final paragraph of the Discussion should be incorporated into the Conclusions section as these paragraphs are somewhat repetitive.

Grammatical errors, wordy style, typos.

Reviewer 3 Report

Abstract: Lines 1-3, check for grammar.

Introduction: Line 20-21, consider to delete the references.

The grammar must be improved.

Use of the word "supposed" in several instances does not read appropriate.

Figure 1 - The caption could be improved to include that four experiments are portrayed. TCS and PTM could be explained.

Figures: what are the error bars?

The choice of words and grammar must be improved.

Round 2

Reviewer 2 Report

The authors have modified the manuscript but have failed to address several important points highlighted in my previous review.
1. The name of the pest insect. FOUR different versions of the common name for the pest appear on the first page, in the title (potato moth), Simple Summary (Guatemalan potato moth), Abstract (tuber moth) and the Introduction (Guatemalan moth). The incorrect name (Guatemalan moth) also persists in the text. Please use one name (I recommend Guatemalan potato moth) throughout the manuscript.
2.  The authors state that they have corrected the degrees of freedom associated with F statistics, but this is not the case. All F statistics need to be accompanied by a numerator and a denominator degrees of freedom, also known as treatment and error degrees of freedom. Please check and correct all F statistics accordingly.
3. Figure 5. The authors have failed to address my question of why the functional response curves in Fig 5 appear to be type II responses, but their statistical models indicate type III response. They clearly state on lines 420-421 that the type III response is sigmoidal. So why isn't Fig 5 (all temperatures) and Fig 6 (27 °C) sigmoidal in nature?
Other points.
Figure 4. Please indicate what "MI" means.
Fig 7. How can the letters reflect the results of a one-way ANOVA when two factors are being compared (parasitoid species, temperature)? Was a two-way ANOVA performed? If not, why?
Table 3. Are these means ± SE?
L438-445. Handling time is important in functional response, but the upper limit to oviposition behavior when many hosts are available is "egg limitation". If the wasps have a fixed number of eggs to lay each day, it makes no difference how many hosts are available, if they exceed the number of eggs the wasp has ready to oviposit.
The additional reference [48] is incomplete (and contains a typo). The correct reference is: Mackauer, M.; Sequeira, R. Patterns of development in insect parasites. In: Parasites and pathogens of insects: Parasites (Vol. 1). Eds. Beckage, N.E., Thompson, S.A., Federici, B.A. Academic Press; San Diego, USA. 1993. pp. 1–23.”

Needs editing.

Round 3

Reviewer 2 Report

The authors have mostly addressed my previous concerns. I detected three issues that remain.

1. Degrees of freedom values are usually separated by a comma, not a slash eg. F = 1.23; df = 2, 16; p = 0.85.

2. The legend to Fig. 7 still mentions one-way rather than two-way ANOVA.

3. There is an error on Line 370 (F6,25? = 406,) and only one value is given for degrees of freedom df = 2. Please correct this and show treatment and error degrees of freedom, as mentioned in Point 1.

Some editing required.
